**Soil respiration database at different time scales in forest ecosystems across China**
Hongru Sun[1,2], Zhenzhu Xu[1], Bingrui Jia[1*]
[1]*State Key Laboratory of Vegetation and Environmental Change, Institute of Botany,*
*Chinese Academy of Sciences, Beijing 100093, China*
[2]*University of Chinese Academy of Sciences, Beijing 100049, China*
[*]Corresponding author:
Bingrui Jia
Institute of Botany, Chinese Academy of Sciences,
20 Nanxincun, Xiangshan, Haidian District, Beijing 100093, China
E-mail: jiabingrui@ibcas.ac.cn
Tel: 86-10-62836289
Fax: 86-10-82595962
**Abstract.** China's forests rank fifth in the world by area and cover a broad climatic
gradient from cold-temperate to tropical zones, and play a key role in the global carbon
cycle. Studies on forest soil respiration ($Rs$) are increasing rapidly in China over the last
two decades, but the resulting $Rs$ data need to be summarized. Here, we compile a
comprehensive database of $Rs$ in China's undisturbed forest ecosystems from literature
published up to December 31, 2018, including monthly $Rs$ and the concurrently measured
soil temperature (N=8317), mean monthly $Rs$ (N=5003), and annual $Rs$ (N=634).
Detailed plot information was also recorded, such as geographical location, climate factors,
stand characteristics, and measurement description. We examine some aspects of the
database – $Rs$ equations fitted with soil temperature, temperature sensitivity ($Q_{10}$),
monthly variations and annual effluxes in cold-temperate, temperate, subtropical and





tropical zones. We hope the database will be used by the science community to provide
a better understanding of carbon cycle in China's forest ecosystems and reduce
uncertainty in evaluating of carbon budget at the large scale. The database is publicly
available at https://www.pangaea.de/tok/788910d8d3ae0a415c7bad2e7025a3f16f042a1b (Sun
et al. 2021).

**Keywords**: Soil carbon flux, Carbon cycle, Temperature sensitivity, Forest, China
**1    Introduction**
Soil respiration ($Rs$) refers to the total amount of $CO_2$ released by undisturbed soil,
including autotrophic respiration and heterotrophic respiration, the former from plant
roots and their microbial symbionts, and the latter from microorganisms decomposing
litter and soil organic matter. As the second-largest terrestrial carbon flux, the recent
estimations of global annual $Rs$ (80–98 Pg C year$^{-1}$) are above ten percent of the
atmospheric carbon pool (750 Pg C) (Bond-Lamberty and Thomson, 2010b**;**
Hashimoto et al., 2015; Raich et al., 2002; Warner et al., 2019), thus accelerating soil
respiration rates with climate warming have a strong potential to influence atmospheric
$CO_2$ levels. It is thus important to understand better soil respiration dynamics and
response to climate changes.

Forest area in China ranks fifth in the world (FAO, 2020) and covers a broad climatic

gradient, including cold-temperate, temperate, subtropical and tropical zones. In China,
most $Rs$ measurements began only after 2001 (Chen et al., 2010), but have rapidly
increased during the last 20 years (Jian et al., 2020). Several studies have summarized
annual $Rs$ in China's forest ecosystems, but with the small samples (e.g., N=50 in
Zheng et al., 2010; N=62 in Chen et al., 2008; N=120 in Zhan et al., 2012; N=139 in
Song et al., 2014). Yu et al. (2010) established a geostatistical model with a total of 390



monthly *Rs* data from different ecosystems in China. With 1782 monthly *Rs* in forest
ecosystems across China, Jian et al. (2020) analyzed the spatial patterns and temporal
trends from 1961 to 2014. However, amounts of *Rs* data are still unexploited, because
they were only displayed in the forms of monthly dynamics in the original papers' figures.
*Rs* data at a subannual timescales are important for upscaling global *Rs* (Jian et al.,
2018), which may derive different conclusions and deserve further exploration (Huang
et al., 2020).

The lack of the large-scale and observation-driven *Rs* data is a main constraining factor

in quantifying regional- to global-scale carbon bugedt (Bond-Lamberty and Thomson,
2010a; Rayner et al., 2005). *Rs* data and concurrently measured temperature thus
provide not only a solid base to understand the critical factors influencing *Rs*, but the
opportunity to better simulate *Rs* at the large scale. We attempted to compile a complete
forest *Rs* database at different temporal scales in China.
**2    Data and methods**
**2.1 Data sources**
The terms of "soil respiration", "soil carbon (or $CO_2$) efflux", or "soil carbon (or $CO_2$)
emission" were searched from publications before 2018 in the China Knowledge
Resource Integrated Database (http://www.cnki.net/), China Science and Technology
Journal Database (http://www.cqvip.com), ScienceDirect (http://www.sciencedirect.
com/), ISI Web of Science (http://isiknowledge.com/), and Springer Link
(http://link.springer.com/). Means, minimums and maximums of soil respiration during
the observation periods were usually given in these published studies, and monthly
patterns of soil respiration rates and the corresponding temperature were frequently
shown with figures. WEBPLOTDIGITIZER, a graphic digitizing software, was used to
take data from figures when values were not reported in the text (Burda et al., 2017).

**2.2 Data collection criteria**


The following criteria were used to ensure data consistency and accuracy: i) *Rs* was
measured in the field without obvious disturbances or manipulation experiments, e.g.
fire, cutting, nitrogen addition treatments, etc. ii) Forested swamps and commercial
plantations (e.g. orchard, rubber, etc.) were not examined. iii) *Rs* was measured either
by static chamber/gas chromatography (GC) or by dynamic chamber/infrared gas
analyzers (IRGA, model Li-6400, Li-8100, Li-8150 (LI-COR Inc., Lincoln, Nebraska,
USA)), which are the most popular methods and provide methodological consistency
(Sun et al., 2020; Wang et al., 2011; Yang et al., 2018; Zheng et al., 2010). Moreover,
the data has been carefully cross-checked by the authors and from different sources.
Based on these criteria, a total of 10288 monthly soil respiration data and 634 annual
soil respiration data were assembled from 568 publications. The dataset covers 28
provinces in China (18.61–52.86° N, 84.91–129.08° E) (Fig. 1). Meanwhile, the related
information was recorded, including geographical location (province, study site, latitude,
longitude and elevation), climate (mean annual temperature and mean annual
precipitation), stand description (forest type, origin, age, density, mean tree height and
diameter at breast height), measurement regime (method, time, frequency, collar area,
height and numbers) (Table 1). This forest region encompasses a large gradient of climate
regimes, mean annual temperature ranging from -5.4 to 23.8 °C and mean annual
precipitation ranging from 105 to 3000 mm.

**2.3 Data verification**


In this study, most of the *Rs* data (~82%) and the concurrent soil temperature at 5 cm
depth ($T_5$) and/or 10 cm depth ($T_{10}$) were extracted with WEBPLOTDIGITIZER, others
(e.g., minimum, maximum) were usually given in the original papers. To verify the



accuracy of the digital software, the means ($Rs$, $T_5$, $T_{10}$) averaged from the extracted data
were compared with the corresponding means directly given in the original papers (Fig.
S1). The coefficients of determination ($R^2$) were all larger than 0.99, indicating that the
accuracy of WEBPLOTDIGITIZER is excellent.
**2.4 Monthly and annual soil respiration calculation**
Long-term continuous $Rs$ could be monitored with Li-8100 or Li-8150, but there are few
published studies of such continuous data (Bond-Lamberty et al., 2020; Tu et al., 2015;
Wu et al., 2014; Yu et al., 2011). The typical days were usually selected to calculate mean
monthly $Rs$ and the observation frequency was 1–12 days per month—high during the
growing season, but low in winter. $Rs$ was measured throughout the day (16%) or at
representative time, e.g., 9:00 a.m.–11:00 a.m. (45%), 9:00 a.m.–12:00 a.m. (22%), etc.,
which had been validated to be close to the diurnal mean value (Xu and Qi, 2001; Yan et
al., 2006; Yang et al., 2018; Yao et al., 2011; You et al., 2013; Zheng et al., 2010). Annual
soil carbon efflux was integrated with soil respiration model (i.e. integration method) or
interpolated the average soil respiration rate between sampling dates (i.e. interpolation
method) (Shi et al., 2014). Finally, monthly $Rs$ and annual soil carbon efflux were
converted to the common unit of $\mu$mol $CO_2$ m$^{-2}$ s$^{-1}$ and g C m$^{-2}$ year$^{-1}$, respectively
(Bond-Lamberty and Thomson, 2010a).
**3    Results**
**3.1 Relationship between soil respiration rate and soil temperature**
Temperature is often the main factor determining soil respiration rates. There were
6341 and 2878 samples of paired $Rs$ & $T_5$ and $Rs$ & $T_{10}$ in the database, respectively.
There were significantly exponential relationships of $Rs$ with $T_5$ and $T_{10}$ in forest
ecosystems across China, which could explain about 48% and 52% of the $Rs$ variations,





respectively (Fig. S2). The exponential correlations were all significant in four climatic
zones, and the coefficients of determination for tropical ecosystems ($R^2$=0.225–0.291)
were smaller than those in other three zones ($R^2$=0.516–0.934) (Fig. 2).
Temperature sensitivity ($Q_{10}$) is defined as the factor by which $Rs$ is multiplied when
temperature increases by 10 °C (Davidson and Janssens, 2006; Lloyd and Taylor, 1994).
$Q_{10}$ could be calculated with the exponential equations between $Rs$ and soil temperature.
At the national scale, the $Q_{10}$ values in China's forest ecosystems from $T_5$ (-16.51–
33.58 °C) and $T_{10}$ (-16.40–33.46 °C) were 2.05 and 2.17, respectively. The $Q_{10}$ was the
largest in cold-temperate zone ($T_5$: 3.74 & $T_{10}$: 3.32), secondary in temperate zone ($T_5$:
2.69 & $T_{10}$: 3.00), and the smallest in subtropical zone ($T_5$: 2.15 & $T_{10}$: 2.20) and
tropical zone ($T_5$: 2.28 & $T_{10}$: 1.63).
**3.2 Monthly dynamics of soil respiration**
Monthly $Rs$ appeared as a single-peak curve (Fig. 3). The largest values occurred in
August (4.18–4.36 µmol m$^{-2}$ s$^{-1}$) in cold-temperate and temperate zones, larger than
the largest values in July (3.58–3.83 µmol m$^{-2}$ s$^{-1}$) in subtropical and tropical zones.
The lowest values occurred in January in cold-temperate (0.20 µmol m$^{-2}$ s$^{-1}$), temperate
(0.49 µmol m$^{-2}$ s$^{-1}$), subtropical (1.10 µmol m$^{-2}$ s$^{-1}$) and tropical zones (1.62 µmol m$^{-}$
$^2$ s$^{-1}$). Monthly variations were largest in cold-temperate and temperate zones,
secondary in subtropical zone, and smallest in tropical zone.
Annual mean $Rs$ in January–December from low to high was cold-temperate (1.63
µmol m$^{-2}$ s$^{-1}$), temperate (1.93 µmol m$^{-2}$ s$^{-1}$), subtropical (2.47 µmol m$^{-2}$ s$^{-1}$) and
tropical zones (2.57 µmol m$^{-2}$ s$^{-1}$). Meanwhile, annual soil carbon emissions were
calculated with the annual mean $Rs$: 621.91 g C m$^{-2}$ yr$^{-1}$ in cold-temperate zone, 733.31
g C m$^{-2}$ yr$^{-1}$ in temperate zone, 937.15 g C m$^{-2}$ yr$^{-1}$ in subtropical zone, and 973.35 g C



m$^{-2}$ yr$^{-1}$ in tropical zone. Soil carbon emissions in growing season (May–October) and
winter (November–April) accounted for 85% and 15% in cold-temperate zone, 80%
and 20% in temperate zone, 69% and 31% in subtropical zone, 61% and 39% in tropical
zone. Subtropical and tropical zones still keep high soil respiration rates in winter,
which is the main source of their larger annual soil carbon emissions.
**3.3 Annual soil carbon effluxes**
There were 634 annual soil carbon effluxes, and most of the observations were
conducted in subtropical zone (61%) and temperate zone (32%) (Fig. 4). Mean annual
soil carbon emission was 851.88 g C m$^{-2}$ yr$^{-1}$ in China's forest ecosystems, ranging
from 260.10 g C m$^{-2}$ yr$^{-1}$ to 2058.00 g C m$^{-2}$ yr$^{-1}$. Mean annual soil carbon emissions in
tropical, subtropical, temperate and cold-temperate zones were 1042.01, 928.91,
697.85 and 684.29 g C m$^{-2}$ yr$^{-1}$, respectively. The former two was significantly higher
than the latter two, but the differences were not significant between tropical and
subtropical zones, and between temperate and cold-temperate zones. The differences
were not significant for EBF, ENF and DNF among different climate zones. DBF in
temperate and subtropical zones was similar (~750.00 g C m$^{-2}$ yr$^{-1}$), both of which were
larger than that in cold-temperate zone (284.20 g C m$^{-2}$ yr$^{-1}$). MF in subtropical zone
(977.35 g C m$^{-2}$ yr$^{-1}$) had significantly higher emissions than that in temperate zone
(733.44 g C m$^{-2}$ yr$^{-1}$).
Evergreen forests were usually larger than deciduous ones in the same climatic zone,
for example, ENF (866.98 g C m$^{-2}$ yr$^{-1}$) and DNF (734.56 g C m$^{-2}$ yr$^{-1}$) in cold-
temperate zone, ENF (699.96 g C m$^{-2}$ yr$^{-1}$) and DNF (555.15 g C m$^{-2}$ yr$^{-1}$) in temperate





zone, EBF (1073.50 g C m$^{-2}$ yr$^{-1}$) and DBF (755.41 g C m$^{-2}$ yr$^{-1}$) in subtropical zone.
Broad-leaved forests showed significantly larger annual fluxes than coniferous forests
in temperate zone (DBF: 748.59 g C m$^{-2}$ yr$^{-1}$ vs. DNF: 555.15 g C m$^{-2}$ yr$^{-1}$) and
subtropical zone one (EBF: 1073.50 g C m$^{-2}$ yr$^{-1}$ vs. ENF: 717.50 g C m$^{-2}$ yr$^{-1}$).
However, DNF (734.56 g C m$^{-2}$ yr$^{-1}$) was larger than DBF (284.20 g C m$^{-2}$ yr$^{-1}$) in cold-
temperate zone, which was from high-latitude Great Xing'an Mountains (~51° N) and
high-altitude Gongga Mountain (2800-2950 m), respectively. Additionally, bamboo is
a special type in subtropical areas, exhibiting the highest soil carbon emissions
(1133.55 g C m$^{-2}$ yr$^{-1}$).
**4   Discussion**
**4.1 Temperature sensitivity ($Q_{10}$) of soil respiration**
$Q_{10}$ is a key parameter in modelling the effects of climate warming on soil carbon
release. The $Q_{10}$ calculated with the exponential equations of $T_5$ and $T_{10}$ were 2.05 and
2.17 at the national scale (Fig. S2), which were lower than the averaged $Q_{10}$ from
different studies in the syntheses of China's forest ecosystems ($T_5$: 2.28–2.51 and $T_{10}$:
2.74–3.00, Peng et al., 2009; Song et al., 2014; Xu et al., 2015; Zheng et al., 2009) and
global forest ecosystems ($T_5$: 2.55–2.70 and $T_{10}$: 3.01–3.31, Wang et al., 2010 a; b).
Our results were close to the $Q_{10}$ of 2 commonly used in many biogeochemical models
(e.g., Cox et al., 2000; Sampson et al., 2007) and the mean $Q_{10}$ of 2.11 estimated with
inverse modeling in forest soils across China (Zhou et al., 2009).
Temperature was the most important limiting factor for soil microbial activity and
root growth in cold regions, thus, *Rs* was more sensitive to temperature changes (Lloyd
and Taylor, 1994; Peng et al., 2009; Zheng et al., 2009; Zheng et al., 2020). The $Q_{10}$



increased from tropical zone to cold-temperate zone in this study, and varied from 1.63
to 3.74. The correlations between *Rs* and soil temperature were lowest in tropical zone
($R^2$=0.225–0.291, Fig. 2d). The difference of the mean *Rs* between tropical moist
forests (1260 g C m$^{-2}$ yr$^{-1}$) and tropical dry forests (673 g C m$^{-2}$ yr$^{-1}$) was about 2-fold
(Raich and Schlesinger, 1992), indicating that soil moisture might play more important
roles.

**4.2 Comparisons of monthly and annual soil carbon effluxes**

The lowest monthly *Rs* occurred in January, and the largest values occurred in August
in cold-temperate and temperate zones and in July in subtropical and tropical zones
(Fig. 3). Similarly, monthly *Rs* of global terrestrial ecosystems reached their minima in
February and peaked in July and August (Hashimoto et al., 2015; Raich et al., 2002).
Due to the limitation of low temperature, winter observations of *Rs* were relatively
fewer in the cold-temperate and temperate zones. The *Rs* in winter (November–April)
was usually assumed to account for 20% of the total annual *Rs* (Geng et al., 2017; Yang
and Wang, 2005), which was in agreement with the proportion in temperate zone, but
greater than 15% in cold-temperate zone.
Annual soil carbon emission had been synthesized in forest ecosystems across China,
and the mean was 745.34 g C m$^{-2}$ yr$^{-1}$ (Zheng et al., 2010), 764.11 g C m$^{-2}$ yr$^{-1}$ (Zhan
et al., 2012), 917.73 g C m$^{-2}$ yr$^{-1}$ (Song et al., 2014) and 975.50 g C m$^{-2}$ yr$^{-1}$ (Chen et
al., 2008), and the mean of 851.88 g C m$^{-2}$ yr$^{-1}$ in the present study was in the mid-
range. The mean annual *Rs* in China's forest ecosystems was slightly lower than the
mean *Rs* of 990.00 g C m$^{-2}$ yr$^{-1}$ in global forest ecosystems (Chen et al., 2010). Warner
et al. (2019) modelled global *Rs* and found that the smallest and greatest annual soil
carbon emissions were in DNF (Mean=344.10 g C m$^{-2}$ yr$^{-1}$) and EBF (Mean=1310.47
g C m$^{-2}$ yr$^{-1}$), respectively. Compared with the predicted annual *Rs*, DNF in cold-



temperate (Mean=734.56 g C m$^{-2}$ yr$^{-1}$) and temperate zones (Mean= 555.15 g C m$^{-2}$ yr$^{-}$
$^{1}$) had larger values, but those of EBF in subtropical (Mean=1073.50 g C m$^{-2}$ yr$^{-1}$) and
tropical zones (Mean=1065.09 g C m$^{-2}$ yr$^{-1}$) were lower (Fig. 4).
Mean annual soil carbon emissions from 634 annual $Rs$ and 5003 mean monthly $Rs$
were 684.29 and 621.91 g C m$^{-2}$ yr$^{-1}$ in cold-temperate zone, 697.85 and 733.31 g C m$^{-}$
$^{2}$ yr$^{-1}$ in temperate zone, 928.91 and 937.15 g C m$^{-2}$ yr$^{-1}$ in subtropical zone, and
1042.01 and 973.35 g C m$^{-2}$ yr$^{-1}$ in tropical zone (Fig. 4 and Fig. 3). The differences
between the directly averaged annual $Rs$ and the accumulative mean monthly $Rs$ were
small in four climatic zones, ranging from -8.24 g C m$^{-2}$ yr$^{-1}$ to 68.66 g C m$^{-2}$ yr$^{-1}$. Mean
annual soil carbon emissions in temperate, subtropical and tropical ecosystems were
745 g C m$^{-2}$ yr$^{-1}$, 776 g C m$^{-2}$ yr$^{-1}$ and 1286 g C m$^{-2}$ yr$^{-1}$ at the global scale, respectively
(Bond-Lamberty and Thomson, 2010a), which were comparable with our results.
**4.3 Improvements of the database**
The common measurement methods were selected, including Li-6400, Li-8100, Li-
8150 and gas chromatography, which had been proved to be consistent (Wang et al.,
2011; Yang et al., 2018; Zheng et al., 2010). The sample sizes of annual $Rs$ were 50–
139 (Chen et al., 2008; Song et al., 2014; Zhan et al., 2012; Zheng et al, 2010) and 634
in the current study, and increased above 4-fold. The global soil respiration database
(SRDB-V5) collected 523 undisturbed annual $Rs$ in China's forest ecosystems (Jian et
al., 2021), but all methods were included, e.g. alkali absorption, gas chromatography
and various infrared gas analyzers. Alkali absorption method could underestimate $Rs$
(Chen et al., 2008; Jian et al., 2020). The total samples of mean monthly $Rs$ were 5003,
which was much larger than the other database's monthly samples of 1782 in China's
forest ecosystems (Jian et al., 2020; Steele and Jian, 2018). Additionally, we extended
the database with the digital software (WEBPLOTDIGITIZER) from the monthly

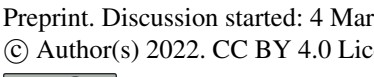



dynamics figures of the original papers, including the paired $Rs$ & $T_5$ (N=6341) and $Rs$
& $T_{10}$ (N=2878). Predicting soil respiration from soil temperature has gained extensive
acceptance (Shi et al., 2014; Song et al., 2014; Sun et al., 2020). These data could be
used to establish the large-scale soil respiration equation and acquire the key parameters
of carbon cycle. Compared with the above-mentioned monthly or annual databases, this
study collected all available $Rs$ data at different time scales. Bamboo forests were
seldom considered in the previous databases (Chen et al., 2008; Steele and Jian, 2018;
Zhan et al., 2012; Zheng et al, 2010), which exhibited the highest soil carbon emissions
(Mean=1133.55 g C m$^{-2}$ yr$^{-1}$, Fig. 4). With the area increasing at a high rate of 3.1%
per year (Song et al., 2017), bamboo forests would play an important role in regional
and even national carbon cycle.
**5   Data availability**
The soil respiration database in China's forest ecosystems used to produce the results
in this study is free to the public for scientific purposes and can be downloaded at
https://www.pangaea.de/tok/788910d8d3ae0a415c7bad2e7025a3f16f042a1b (Sun et
al. 2021).
**6   Conclusions**
In this study, we collected in situ $Rs$ measurements with common infrared gas analyzers
(i.e. Li-6400, Li-8100, Li-8150) or gas chromatography to assemble a comprehensive
and uniform database of China's forest ecosystems at different time scales. Besides the
$Rs$ data directly given in the original papers, the monthly patterns of $Rs$ and the
concurrently measured soil temperature at 5 cm and/or 10 cm depth in the figures were
digitized. Meanwhile, we have made a preliminary analysis of the data. The results
showed that soil temperature could explain 22.5%–93.4% of the $Rs$ variations.



Temperature sensitivity ($Q_{10}$) was about 2.05–2.17 at the national scale, increasing
from 1.63 in tropical zone to 3.74 in cold-temperate zone. Monthly *Rs* showed a single-
peak curve, and the largest values occurred in August (4.18–4.36 µmol m$^{-2}$ s$^{-1}$) in cold-
temperate and temperate zones, larger than the largest values in July (3.58–3.83 µmol
m$^{-2}$ s$^{-1}$) in subtropical and tropical zones. Mean annual soil carbon emissions decreased
from tropical (1042.01 g C m$^{-2}$ yr$^{-1}$), subtropical (928.91 g C m$^{-2}$ yr$^{-1}$), temperate
(697.85 g C m$^{-2}$ yr$^{-1}$) to cold-temperate zones (684.29 g C m$^{-2}$ yr$^{-1}$). This study provides
basic data and scientific basis for quantitative evaluation of soil carbon emissions from
forest ecosystems in China.
**Author contributions**. BJ designed the soil respiration database and searched the
papers until 2018. HS and BJ collected and digitized soil respiration data and compiled
the associated information. HS and BJ prepared the manuscript. ZX provided many
useful suggestions and reviewed the paper.
**Competing interests**. The authors declare that they have no conflict of interest.
**Acknowledgements.** We are grateful to the scientists who contributed their work to
our database. We thank Ben Bond-Lamberty for the constructive comments and
improvements to this manuscript. This work was supported by the National Natural
Science Foundation of China (32071592) and the National Key Research and
Development Program of China (2017YFC0503906).

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





**Table 1.** Variable information of soil respiration database in China's forest ecosystems,
available at https://www.pangaea.de/tok/788910d8d3ae0a415c7bad2e7025a3f16f042a1b. N/A
refers to values that are not applicable.

| Column | Description | Unit | Number | Range |
|---|---|---|---|---|
| ID | Unique identification number of each record | N/A | 11297 | 1–11297 |
| Province | Province location of study site | N/A | 28 | N/A |
| Study site | Name of study site | N/A | 155 | N/A |
| Latitude | Latitude (N) of study site | ° | 988 | 18.61–52.86 |
| Longitude | Longitude (E) of study site | ° | 988 | 84.91–129.08 |
| Altitude | Altitude of study site | m | 988 | 7–4200 |
| MAT | Mean annual temperature | °C | 988 | -5.4–23.8 |
| MAP | Mean annual precipitation | mm | 988 | 105–3000 |
| Forest type | Forest community characterized by the dominant tree species, or the ecological similarities (e.g. life form and biotope) | N/A | 180 | N/A |
| Origin | Stand origin was classified into planted and natural (i.e. secondary, primary) forests | N/A | 4 | N/A |
| Age | Stand age | years | 769 | 2– ~400 |
| DBH | Mean diameter at breast height | cm | 610 | 2.40–51.96 |
| $H_{tree}$ | Mean tree height | m | 538 | 2.50–48.00 |
| Density | Stem density and/or canopy coverage | trees ha$^{-1}$ | 548 | 209–17000,0.23–0.98 |
| Instrument | Measurement instrument of $Rs$, i.e. gas chromatography, infrared gas analyzers (Li-6400, Li-8100, Li-8150) | N/A | 4 | N/A |
| Time | Observation time of $Rs$ | Hour: Minute | 749 | 0:00–23:00 |
| Frequency | Observation frequency of $Rs$, i.e. days per month | days | 961 | 0.5–31 |
| Area | Observation area of $Rs$, i.e. area of soil collar or base | cm$^2$ | 976 | 50–2500 |
| Height | Height of soil collar or chamber | cm | 828 | 4–50 |
| Replication | Numbers of soil collar or chamber | N/A | 968 | 1–768 |
| Month | Observation month of $Rs$ | Month, Year | 10288 | Jan.,2000–Mar.,2018 |
| $Rs$ | Soil respiration rate | µmol m$^{-2}$ s$^{-1}$ | 10288 | 0.01–11.84 |
| $T_5$ | Soil temperature at 5cm depth concurrently measured with $R_S$ | °C | 6341 | -16.51–33.58 |
| $T_{10}$ | Soil temperature at 10cm depth concurrently measured with $R_S$ | °C | 2878 | -16.40–33.46 |
| Mode | The ways to obtain $Rs$ data, 1: extracted with WEB PLOTDIGITIZER, 2: directly given in the original study | N/A | 2 | 1–2 |
| Period | Period of annual soil carbon efflux | Month, Year | 631 | Jan.,2001–Mar.,2018 |
| Annual $Rs$ | Annual soil carbon efflux | g C m$^{-2}$ year$^{-1}$ | 634 | 260.10–2058.00 |
| Method | Method to calculate annual soil carbon efflux, i.e. integration method and/or interpolation method | N/A | 3 | N/A |
| Reference | Data sources | N/A | 568 | N/A |




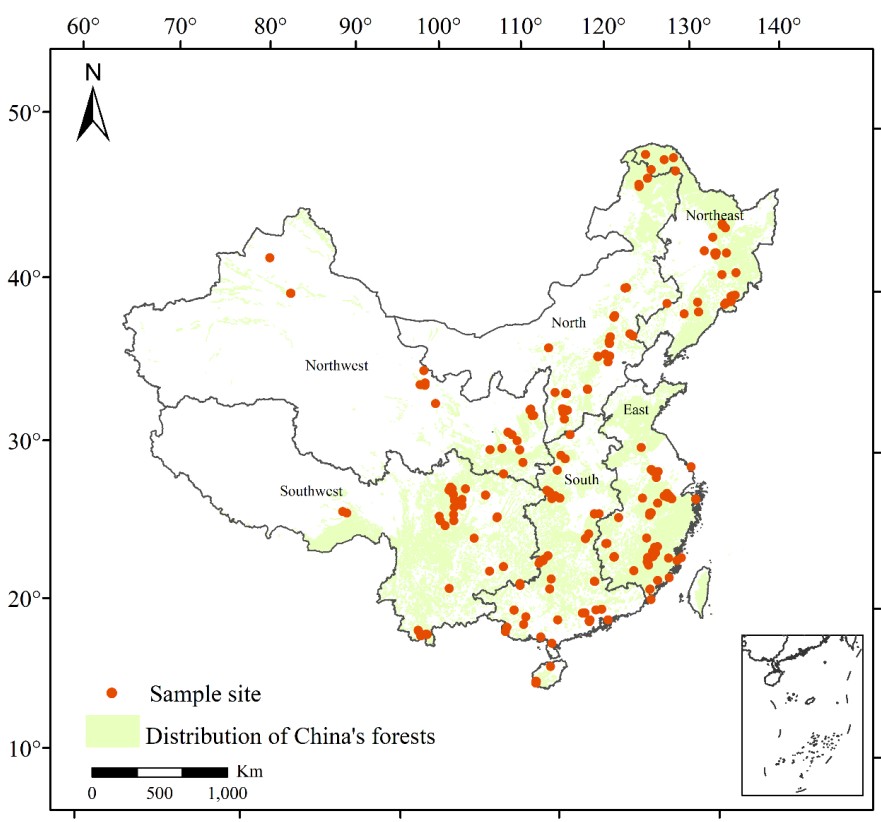


**Figure 1.** Distribution of study sites used to develop the forest soil respiration database

in China.

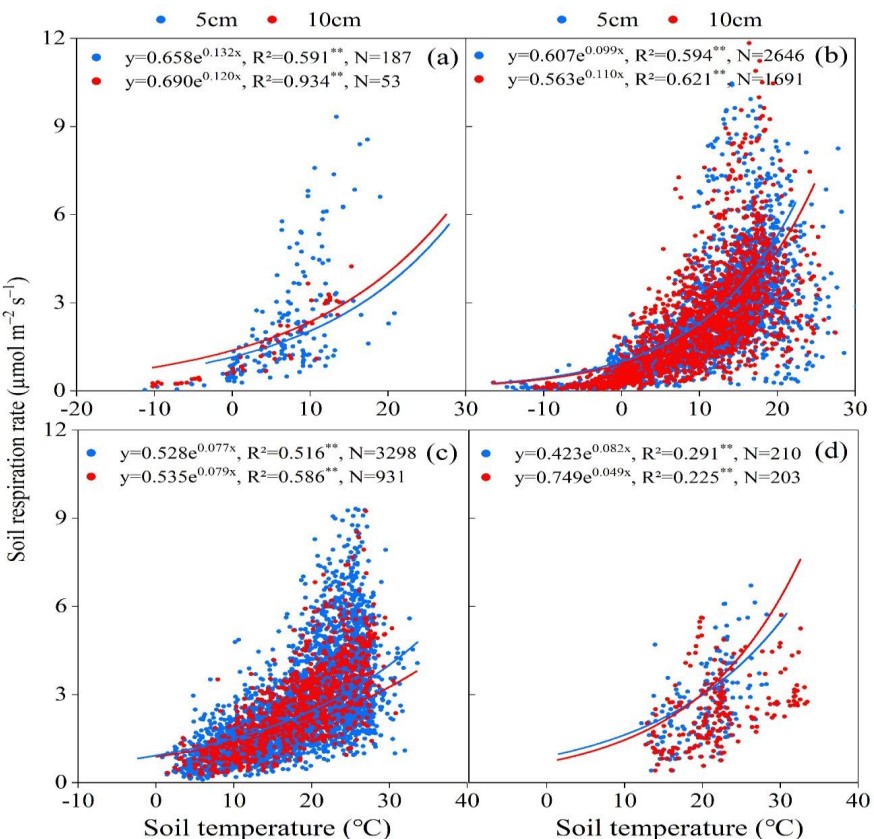

**Figure 2.** Exponential relationships of forest soil respiration rates with soil temperature
at 5 cm depth and 10 cm depth in cold-temperate (a), temperate (b), subtropical (c) and
tropical zones (d). *P* value below 0.01 was described by [**].

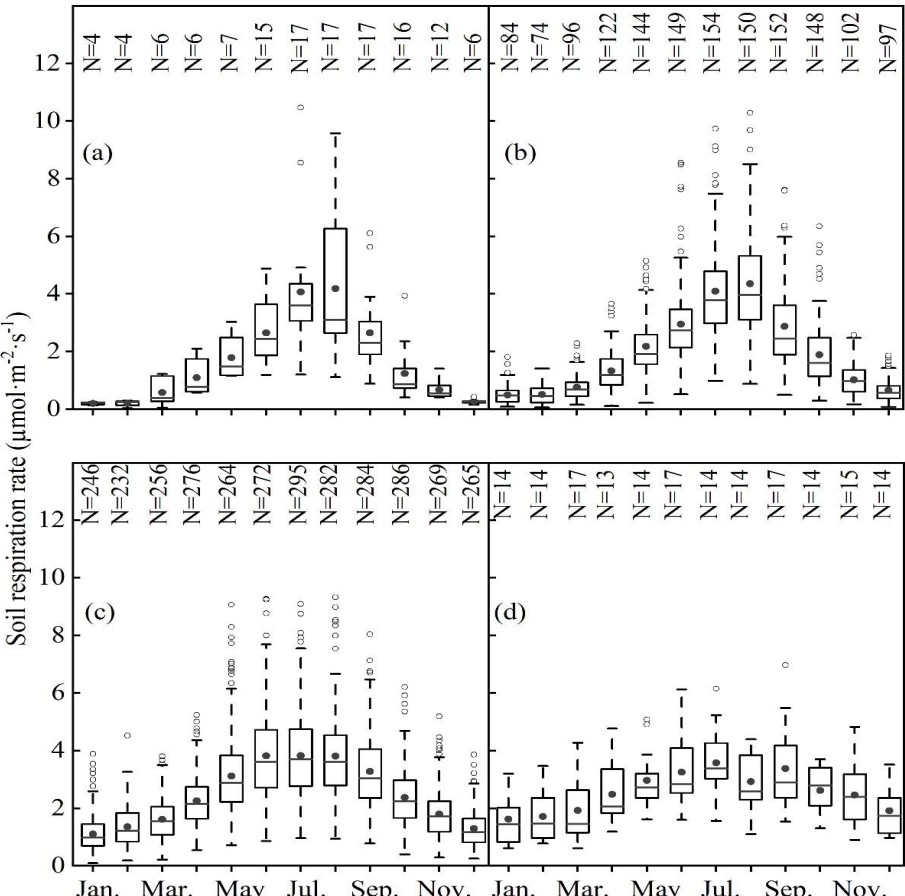

**Figure 3.** Monthly patterns of forest soil respiration rates in cold-temperate (a), temperate (b), subtropical (c) and tropical zones (d). Solid circle: mean value; Solid horizontal line: median; Box: 25th to 75th percentiles; Whisker: 1.5 times interquartile range; Open circle: data points beyond the whiskers. The samples per month were listed in the upper part of the figure.

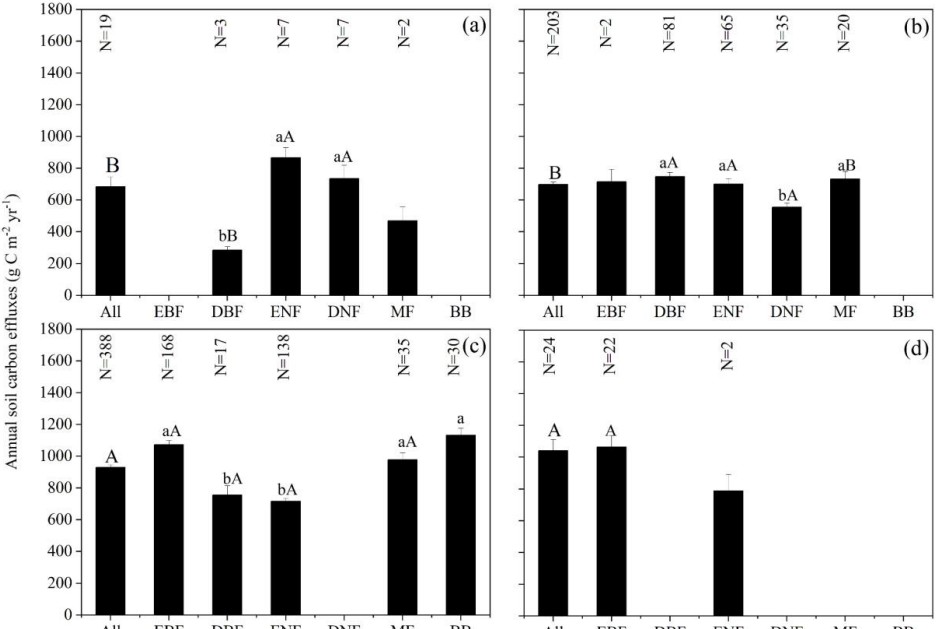

**Figure 4.** Comparisons of annual soil carbon effluxes (mean ±standard error) among different forest types across China in cold-temperate (a), temperate (b), subtropical (c) and tropical zones (d). Lowercase letters are the comparisons of different forest types in each climatic zone, while capital letters are the comparisons of the same forest type in different climatic zones. The samples were listed in the upper part of the figure, and the samples larger than 3 were compared. EBF: evergreen broadleaf forest, DBF: deciduous broadleaf forest, ENF: evergreen needleleaf forest, DNF: deciduous needleleaf forest, MF: broadleaf and needleleaf mixed forest and BB: Bamboo forest.