# Peer review of "A compiled soil respiration dataset at different time scales for forest ecosystems"

_Earth System Science Data, 2021_

## Author Comment (AC1)

N

Annual soil carbon efflux
g C m$^{-2}$ yr$^{-1}$

- 260.10 - 666.30
- 666.30 - 961.00
- 961.00 - 1306.00
- 1306.00 - 2058.00

1512.46
434.58

Nansha Islands

---

## Author Response (AR1)

Title: Soil respiration database at different time scales in forest ecosystems across China

**Dear Editor and Referees:**

Thank you very much for your kind consideration and help to our manuscript! According to your suggestions, we revised our manuscript. All the modifications were listed as follows.

**Referee #1:**

**Comment:** Soil respiration is an important indicator for a wide range of applications, especially those related to evaluating carbon cycle regionally or globally. The authors did a great job for collecting a total of 10288 monthly and 634 annual soil respiration data from 568 publications. I enjoyed reading this well-reasoned and well written study. In a certain extent, it is helpful for building robustness of this dataset for comparison between that from reference and software. However, it still requires substantive effort for the following reason:

1. Authors mentioned Yu et al. (2010) established a geostatistical model with 390 monthly data and Jian et al. (2020) analyzed the spatial patterns and temporal trends with 1782 monthly data. The authors need to justify the importance of their research in comparison with these researches. For example, using any quantitative method to address advantage of their dataset. It just looks like a supplement for the research mentioned above right now.

**Response:** The importance of the dataset was discussed in the section "**4.3 Improvements of the dataset**" in Lines 260-290, mainly including the following four aspects: 1) the large increase of samples in annual Rs (N=634) and monthly mean Rs (N=5003), 2) the concurrently measured Rs & $T_5$ (N=6341) and Rs & $T_{10}$ (N=2878), which were extracted from the figures in the original papers with the digital software (WEBPLOTDIGITIZER) and not supplied in the previous datasets, 3) the consistency of the selected measurement methods (Li-6400, Li-8100, Li-8150 and gas chromatography), 4) Bamboo forests included, which were seldom considered in the previous datasets.

**Comment:** 2. What is the difference of soil respiration among different equipment and method? How does these affect the robustness of the dataset?

**Response:** "The common measurement methods were selected, including Li-6400, Li-8100, Li-8150 and gas chromatography, which had been proved to be consistent" **was revised to** "Rs measurements were mainly from Li-8100 (47%) and Li-6400 (33%), secondary from gas chromatography (18%), and Li-8150 only accounted for 2%. The differences of the four common measurement methods had been proved to be small (~10%)" in Lines 261-263.

**Comment:** 3. Authors need to add more information for 'cross-checked'. It will be helpful to add one table or figure to address the different sources.

**Response:** There are some data from the same authors and different sources (e.g. master or Ph. D. dissertation and journal article). Here, "cross-checked" means the data from different sources was checked. To avoid to misunderstand, "Moreover, the data has been carefully cross-checked by the authors and from different sources." **was revised to** "Moreover, the data from the same authors and different sources (e.g. master or Ph. D. dissertation and journal article) has been carefully cross-checked and supplemented." in Lines 106-108.

**Comment:** 4. In Table 1, it is confused that the number of latitude and longitude are more than Study site. I only found 251, 122 and 180 different values for latitude, MAT and MAP, respectively. It needs more clarification for this table.

**Response:** Thanks for your reminder. In Table 1, we showed the number of the different study sites, but the numbers of the latitude, longitude, altitude, MAT and MAP were all. Thus, to keep the consistency, the numbers of the different latitude, longitude, altitude, MAT and MAP were listed in Table 1, i.e. 208, 218, 329, 122 and 180, respectively.

**Comment:** 5. What are the patterns for soil respiration along MAT and MAP?

**Response:** The patterns for annual soil respiration along MAT and MAP **were supplemented** in Figure S3.

**Comment:** 6. I strongly recommend that the authors rasterize this dataset to about 10 km resolution. You can exclude the Northwest and Southwest part of China which did not covered with much forests. Then, it would be more compatible to do analysis with spatial climate data.

**Response:** Thank you for your suggestion. This is a good advice. We rasterized this dataset to 1 km resolution in the following figure. Yet compared with the large forest area of China ($188 \times 10^4 \, km^2$), the small samples (N=634) and their unbanalanced distributions would increase the uncertainties. Data coverage is a challenge for larger-scale extrapolation, and the simple interpolation analysis based on this data may be inaccurate. In order to make a more precisely map, in the future study, we are going to use more reasonable methods like machine learning, and have climate and soil variables included as predictors.

[Figure]

Annual soil carbon efflux
g C m$^{-2}$ yr$^{-1}$

· 260.10 - 666.30
· 666.30 - 961.00
● 961.00 - 1306.00
● 1306.00 - 2058.00

1512.46
434.58

Nansha Islands

**Referee #2:**

**Comment:** Overall: The dataset is a new compilation of soil respiration rates across China, available from a range of literature. The compilation has an impressive number of values collated. Whilst, by nature, the data is not unique on an individual basis due to being extracted from other sources, the compilation as a whole is a unique resource.

The dataset seems like a potentially useful compilation of values in terms of investigating climate change in China and globally. The values cover a broad range of climatic zones. The work attempts to standardise measurements in terms of different temporal scales, and explains the methods for doing this, which is commendable.

**Response:** Thank you very much for your kindly comments and encouragements. We are trying to establish a comprehensive and standardized forest soil respiration database across China, which could be useful in the related studies on carbon cycle and climate change in China and globally.

**Comment:** The discussion uses the data to link soil temperature to soil respiration, and thus climate change, and suggests further work could be carried out in relation to soil moisture.

**Response:** Indeed, soil moisture is an important factor. Many soil moisture datasets have been

developed (e.g., Chen et al., 2021; Guevara et al., 2021; Meng et al., 2021; Wang et al., 2021), we are going to carry out the related analyses in the future study.

Chen Y, Feng X, Fu B. 2021. An improved global remote-sensing-based surface soil moisture (RSSSM) dataset covering 2003-2018. Earth System Science Data, 13: 1-31.

Guevara M, Taufer M, Vargas R. 2021. Gap-free global annual soil moisture: 15 km grids for 1991-2018. Earth System Science Data, 13: 1711-1735.

Meng X, Mao K, Meng F, et al. 2021. A fine-resolution soil moisture dataset for China in 2002-2018. Earth System Science Data, 13: 3239-3261.

Wang Y, Mao J, Jin M, et al. 2021. Development of observation-based global multi-layer soil moisture products for 1970 to 2016. Earth System Science Data, 13: 4385-4405.

**Comment:** The WEBPLOTDIGITIZER method to extract values from figures seems interesting and an assurance of the quality of these data is given in Section 2.3.

**Response:** Yes, the extracted data with WEBPLOTDIGITIZER were verified in Section 2.3 in Lines 96-106.

**Comment:** Attempts are evident to show data consistency in collection in Section 2.2 in terms of only choosing Rs values measured from undisturbed ground, and in terms of the instruments used. It could perhaps be explained a little more as to the potential differences that could arise by using different equipment, and how this might affect the dataset.

The article overall is succinct, well-structured and clear.

**Response:** "The common measurement methods were selected, including Li-6400, Li-8100, Li-8150 and gas chromatography, which had been proved to be consistent" **was revised to** "Rs measurements were mainly from Li-8100 (47%) and Li-6400 (33%), secondary from gas chromatography (18%), and Li-8150 only accounted for 2%. The differences of the four common measurement methods had been proved to be small (~10%)" in Lines 261-263.

**Comment: Data quality** The dataset is easily accessible via the given identifier.

I would normally expect a non-proprietary format for long-term storage/publication of data − e.g. comma separated values (.csv) rather than Microsoft Excel (.xlsx), for purposes of longevity, and to ensure the maximum number of users are able to open the dataset in freely available software.

**Response:** With your suggestions, the format of the dataset was changed to a non-proprietary data format (.csv) in the repository in Pangaea.

**Comment:** I would have expected the sample information (Province, Study site, Lat/long etc.) to persist for each data point, rather than there being rows of blank information. The assumption is that samples below the first instance of each Province, Study site etc., are the same/related, however, if

you were to re-sort the spreadsheet, you would lose this associated information from the samples with blanks - each row is not 'stand-alone' as it should be.

This also means that it is not clear as to the difference between samples – for example, there are two data points marked with "Aug.,2013" – but what is the difference between the two? There is nothing on the individual rows to explain or describe.

**Response:** With your suggestions, the table was split into two related datasets with the same ID in the repository in Pangaea, one includes the soil respiration and temperature data and the other one the metadata of the sample information for each study.

Thanks for your reminder. Soil respiration was usually measured a few days per month. Means per month were only given in most studies, but a few values per month were all given in some studies. "monthly means or a few values per month" **was supplemented** in Table 1.

**Comment:** Again related to this, there is no way to automatically calculate the means in order to check their accuracy, because there is no field value by which to group the values to create the mean. The mix of data types in one column also precludes this – e.g. a numeric value column, with "NA" (string/character format) for missing values. I would have expected a numeric code to denote "NA", or a separate column containing the "NA".

**Response:** Means of soil respiration rate, soil temperature at 5 cm depth and 10 cm depth in each study were given in the columns of "Rs", "$T_5$" and "$T_{10}$", respectively. In order to calculate conveniently, the missing values (i.e. NA) in the numeric value columns of "Rs", "$T_5$"and "$T_{10}$" **were deleted** in the dataset.

**Comment:** Error estimates are not given in the dataset, although it is not clear as to whether this would be appropriate, based on the data extracted from the sources. Overall errors are presented in Figures 3 & 4.

**Response:** Root Mean Square Error (RMSE) was supplemented in Fig. 2, Fig. S1, Fig. S2, Fig. S3. "standard error" **was added** in Lines 176-202.

**Comment:** Whilst metadata is available in the article under review (e.g. table 1), I would have expected a metadata document (containing field level metadata) to accompany the data download, in addition to the summary given on the Pangaea landing page, which is not very detailed.

**Response:** A readme file was supplemented in the dataset in Pangaea.

**Comment: Specific Issues**    Line 57 – bugedt -> budget

**Response:** "bugedt" **was revised to** "budget" in Line 57.

**Referee #3:**

**Comment:** Based on a thorough review of 568 original research articles and other publications, Sun

et al. compiled a comprehensive soil respiration dataset that covers a wide range of climates, elevations and forest ecosystems across China. The dataset comprises a total of 10288 monthly and 634 annual soil respiration measurements and for some sites also monthly soil temperature measurements (at 5 and/or 10 cm depth). In addition, specific information (geographic location, forest type, mean annual air temperature and precipitation, etc.) are provided for each site. In view of the vast number of independent soil respiration studies from different regions, consistent datasets that summarise the state of the art and present the available data in a common format are of great benefit for the research community as they facilitate, for example, the analysis of spatial variations and temporal trends in soil carbon emission. The manuscript is generally well-written and the entire dataset is made publicly available through the open-access data repository PANGAEA. However, I have some concerns and suggestions that I would like to see addressed before I can recommend the manuscript for publication.

As I am not an expert in the field of soil respiration, my comments focus mainly on the overall content of the manuscript and the structure of the dataset.

**Response:** Thank you very much to take the time to review and improve our manuscript and the dataset.

**Comment: General comments**

Uniqueness of the dataset: A similar dataset as the one presented by Sun et al. has been published by Jian et al. (2020) for different forest ecosystems across China, although it is less comprehensive. I therefore encourage the authors to clearly state the added value of their dataset compared to previous studies.

**Response:** Uniqueness of the dataset was discussed in the section "**4.3 Improvements of the dataset**" in Lines 260-286, for example, the consistency of the measurement methods, the large increase of monthly and annual Rs samples, soil respiration rates and concurrently measured soil temperature extracted from the figures with the digital software (WEBPLOTDIGITIZER), the extension of forest types (including Bamboo forests).

**Comment:** Period of the dataset: I acknowledge the effort of the authors to screen 568 publications and compile all the data, but it would be highly desirable if the dataset could be extended until 2020 so that it would cover a 20-year period (2000-2020), in line with the last two decades of the latest climate reference period (1991-2020). Such a dataset would facilitate the joint analysis of spatio-temporal climate and soil respiration variations (in the context of climate change). Similar to the title for the data repository (Sun et al. 2021), the period of the presented dataset should also be included in the title and abstract of the manuscript. Moreover, it would be helpful to mention the

178    (average) length of the individual time series from the different sites somewhere in the text or to

179    provide a respective figure (e.g. histogram) in the supplements.

180    **Response:** Thanks for your suggestions, it took us three years (from 2019 to now) to establish the

181    dataset, it's an interesting and valuable work, we will go on updating the dataset in the future study.

182    Title "Soil respiration database at different time scales in forest ecosystems across China" **was**

183    **revised to** "A compiled soil respiration dataset at different time scales for forest ecosystems across

184    China from 2000 to 2018".

185    Frequency distribution histogram of the length of the individual time series from the different sites

186    was supplemented in Fig. S4. "Fig. S4 showed the length of the individual time series from the

187    different sites, the high frequencies were 12 months (38%), 6–7 months (20%) and 13–24 months

188    (15%)." **was added** in Lines 280-282.

189    **Comment:** Uncertainties: As mentioned in chapter 2.3, most of the soil temperature and respiration

190    data (82 %) were extracted with the WEBPLOTDIGITIZER. This in an interesting approach that

191    provides a workaround to compile scientific data that are not made publicly available in the original

192    studies. However, I was wondering if the authors of these studies have been contacted to request

193    access to the numeric data or was this not feasible due to the number of studies? Were the data from

194    the 568 studies (many of them non-peer-reviewed) checked manually or automatically to identify

195    potential errors or inconsistencies?

196    **Response:** The database included 568 literatures, it was not feasible to contact all of the authors.

197    The data in figures were extracted by Hongru Sun and checked manually by Bingrui Jia.

198    Additionally, the data from the same authors and different sources (e.g. master or Ph. D. dissertation

199    and journal article) has been carefully cross-checked.

200    **Comment:** The given $R^2$ values of 0.99 for the simple linear regressions (original mean soil

201    respiration data vs. digitised soil respiration data) seem promising, but how does it look like for the

202    monthly data? As a measure for uncertainty, the RMSD or MAE should be provided as well. I am

203    also missing a section in the manuscript that discusses (at least qualitatively) the potential

204    uncertainties originating from the different instruments and experimental setups at the different

205    study sites as well as from the varying time periods of the datasets used for characterising

206    differences between the four climate zones (cold-temperature, temperate, subtropical and tropical).

207    Lastly, are there forest types (e.g. mountain forests) that are potentially under-represented in the

208    dataset (due to a lack of respective studies)? Such a potential bias might affect the numbers

209    provided for the temperature sensitivity of soil respiration and for the annual soil carbon emission

210    originating from forest ecosystems in China. This needs to be to discussed at least briefly.

**Response:** Yes, these were monthly data in Fig. S1, because most of the monthly data were shown with figures and the annual data were directly given in the original papers. Root Mean Square Error (**RMSE**) (also called the root mean square deviation, **RMSD**) was supplemented in Fig. S1.

"Rs measurements were mainly from Li-8100 (47%) and Li-6400 (33%), secondary from gas chromatography (18%), and Li-8150 only accounted for 2%." **was added** in Lines 261-262.

"The spanning years were 2003–2014 in cold-temperate zone, 2000–2018 in temperate zone, 2002–2017 in subtropical zone and 2003–2017 in tropical zone." **was added** in Lines 172-174.

"It's worth noting that the Rs studies were fewer in the regions of latitude larger than 48° (~2%) or elevation higher than 3000 m (~4%). The potentially under-represented forest types might affect the evaluation of temperature sensitivity of soil respiration and annual soil carbon emission at the regional and national scale." **was added** in Lines 287-290.

**Comment:** I would suggest the following modifications for the dataset:

Use a non-proprietary data format (e.g. CSV file) so that the dataset can be easily read by any software.

**Response:** The format of the dataset was changed to a non-proprietary data format (i.e. CSV file) in the repository in Pangaea.

**Comment:** Add a metadata file or readme file that contains all necessary information (e.g. those from Table 1 in the manuscript) so that the dataset can theoretically be used independently of the data paper. Nevertheless, add a reference to the data paper in the metadata/readme file and on the landing page of the repository.

**Response:** A readme file was supplemented in the dataset in Pangaea. There was a reference on the landing page of the repository in Pangaea.

**Comment:** Create a GeoPackage (.gpkg) or Shapefile (.shp) that contains the metadata (coordinates, elevation, study site name, forest type, etc.) for each study site. Include it in the repository so that it can be easily imported in a GIS by potential users for spatial data visualisation and analysis.

**Response:** Thank you for your suggestions. We created an ESRI Map Package (.mpk), the metadata was included in the package and was added in the repository in Pangaea. Potential users for spatial data visualisation and analysis can use this ESRI Map Package.

**Comment:** Add the units (of each column) either in the header or in the metadata/readme file.

**Response:** The units were added in the readme file in the repository in Pangaea.

**Comment:** Column "Month": Use the international date format (ISO 8601) or another common date format that can be easier interpreted by a machine (e.g. 2013-07 instead of Jul.,2013). Replace "Month" by "Date".

**Response:** Column "Month" **was revised to** a common date format (e.g. 2013-07). "Month" **was revised to** "Date".

**Comment:** No need for column "Period" as the necessary information are already provided in column "Month" ("Date").

**Response:** Column "Period" means the period of annual soil carbon efflux and is necessary, because some studies only supplied annual data, but not monthly data or only part of monthly data.

**Comment:** Column "Time": which time is meant here? No time zone provided. It is unclear to which data the time refers. Split into two columns as well: e.g. "Start" and "End".

**Response:** Soil respiration is usually measured a few days per month, here, "Time" means the observation time per day. "Observation time of Rs" **was revised to** "Observation time of Rs per day (Beijing time)" in Table 1. There were 3 observation times (daily, monthly, yearly) in the dataset, splitting into two columns would be more confused.

**Comment:** Remove the tilde in "Age" as this special character is difficult to handle during automatic processing. Alternatively include a column before or after and use a flag (1, 0) to indicate whether the "Age" is measured/precise (e.g. 1) or estimated/approximate (e.g. 0).

**Response:** Age of a natural forest is generally estimated from historical records or dominant tree rings, and the age of a planted forest is defined since planting. Thus, the tilde "~" in column "Age" was deleted, and "estimated from historical records or dominant tree rings in natural forest, defined since planting in planted forest" **was added** in Table 1.

**Comment:** Column "Forest type": would it be possible to use an integer or acronym for each forest type in the database and provide the full name in the metadata/readme file?

**Response:** To give the detailed forest community of the study site, forest type was characterized by the dominant tree species, or the ecological similarities (e.g. life form and biotope). The number of forest type was 180, it was not suitable to substitute with an integer or acronym.

**Comment:** Columns "Rs", "T5", "T10": Better remove "NA" (leave cells empty) and create another column before or after indicating with a flag whether data are available (e.g. 1) or lacking (e.g. 0). The same for the other columns where NA values exist.

**Response:** The missing values (i.e. NA) in the numeric value columns of "Rs", "$T_5$" and "$T_{10}$" **were deleted** in the dataset. A total of 17 columns included "NA" in the dataset. If we created the additional 17 columns, the dataset would become complicated.

**Comment:** Column "Annual Rs": Does this column indeed provides annual averages or rather the mean over the study period? I think it can be deleted as the mean can be easily calculated from the monthly data provided.

**Response:** Column "Annual Rs" is the annual soil carbon efflux (g C m$^{-2}$ year$^{-1}$), not the mean over the study period.

**Comment:** Column "Altitude": Replace "Altitude" by "Elevation".

**Response:** Column "Altitude" **was replaced by** "Elevation".

**Comment:** Although the redundancy may increase the file size of the dataset considerably, I would recommend to copy the metadata (geographic information etc.) into each line (not only in the first row from each site). Otherwise, complications may arise when the dataset is reformatted or analysed. Alternatively, the table could be split into two related datasets. One would include the soil respiration and temperature data and the other one the metadata for each site. An additional ID could be provided for each study site to link the two datasets...

**Response:** With your suggestions, the table was split into two related datasets, one includes the soil respiration and temperature data and the other one the metadata for each study.

**Comment: Specific comments**

Title: Mention the timeframe of the dataset ("2000-2018" or "2000-2020") and replace "database" by "dataset". A database describes a collection of multiple datasets that are generally stored and accessed electronically from a computer system... Title suggestion: "A compiled monthly soil respiration dataset for [various] forest ecosystems across China from 2000 to 2018"

**Response:** Title "Soil respiration database at different time scales in forest ecosystems across China" **was revised to** "A compiled soil respiration dataset at different time scales for forest ecosystems across China from 2000 to 2018".

**Comment:** Line 64-73: Maybe just quote the database here and add the URL (with the access date) in the reference list.

**Response:** The Five databases were searched the related soil respiration studies to compile our dataset, it was more suitable to directly introduce them in the Section "**2.1 Data sources**".

**Comment:** Line 75-83: Indicate the period that has been considered. From 2000 until 2018 I think.

**Response:** "The observation years were from 2000 until 2018." **was added** in Lines 94-95.

**Comment:** Line 85: Do the 568 publications represent 568 study sites or are some data from the same site? Please state the number of considered sites and their geographic and elevational distribution somewhere in the text.

**Response:** Some data from multi-sources in the same site, forest type and author were merged in the dataset. "The dataset covers 28 provinces in China (18.61–52.86 °N, 84.91–129.08 °E)" **was revised to** "There were 155 study sites from 28 provinces in China (18.61–52.86 °N, 84.91–129.08 °E, 7–4200 m)" in Lines 91-92.

**Comment:** Line 86-91: Why have the other provided variables (e.g. mean annual temperature and precipitation as well as elevation) not been included in the analysis? I am aware that an in-depth analysis is beyond the scope of this data paper, but some additional plots (soil respiration vs. elevation, or soil respiration along selected temperature or precipitation transects) would emphasise and showcase potential of this dataset.

**Response:** Thanks for your suggestions. The plots of annual soil carbon effluxes with mean annual temperature and precipitation was added in Figure S3 in supplementary material. "The annual soil carbon effluxes increased with the increasing of mean annual temperature and precipitation at the national scale (Fig. S3). " **was added** in Lines 176-178.

**Comment:** Line 100: In addition to $R^2$, provide the RMSD or the MAE as a measure for uncertainty.

**Response:** "The Root Mean Square Errors (RMSE) of Rs, $T_5$ and $T_{10}$ were 0.09 μmol m$^{-2}$ s$^{-1}$, 0.35 ℃ and 0.44 ℃, respectively" **was added** in Lines 104-105.

**Comment:** Line 105-107: Please clarify whether this sentence describes the procedure in the original studies or how you have modified and analysed the data.

**Response:** This sentence describes the procedure in the original studies, we didn't modify and analyze the data. To avoid to misunderstand, " The typical days were usually selected to calculate mean monthly Rs" **was deleted**.

**Comment:** Line 115: I am missing a brief section in the methods about the (statistical) analysis that had been performed to present the data.

**Response:** The section "2.5 Statistical analysis" **was added** in Lines 122-132.

**Comment:** Line 119: Provide in addition to the total number of paired measurements (6341 and 2878) also the percentage (in parenthesis) with respect to the total number of considered data.

**Response:** "There were 6341 and 2878 samples of paired Rs & $T_5$ and Rs & $T_{10}$ " **was revised to** "The samples of the paired Rs & $T_5$ and Rs & $T_{10}$ were 6341 (69%) and 2878 (31%)" in Lines 135-136.

**Comment:** Line 123-124: Note that $R^2$ is invalid/inappropriate for non-linear regressions! $R^2$ cannot differentiate between "good" and "bad" non-linear models. The standard error of the regression could be used instead.

**Response:** "the coefficients of determination for tropical ecosystems ($R^2$=0.225–0.291) were smaller than those in other three zones ($R^2$=0.516–0.934)" **was revised to** "RMSEs in cold-temperate and temperate zones (1.52–1.67 μmol m$^{-2}$ s$^{-1}$) were larger than those in subtropical and tropical zones (1.04–1.32 μmol m$^{-2}$ s$^{-1}$), except the smallest RMSE from $T_{10}$ in cold-temperate zone (0.42 μmol m$^{-2}$ s$^{-1}$)." in Lines 140-143.

**Comment:** Line 125-132: Uncertainties need to be provided for these values!

**Response:** These $Q_{10}$ values were calculated with the exponential equations between all soil respiration and soil temperature data in each climate zone, but not the means. Additionally, RMSEs of the exponential equations between Rs and soil temperature were given in Figure 2 and Figure S2.

**Comment:** Line 134-150: One big problem I see here is that time series spanning different years were used to determine seasonal and geographic differences in soil respiration. I am aware that this is unavoidable when using a compiled dataset, but the uncertainties originating from this issue should be at least discussed qualitatively. Which period do the data cover that were used to compute these values (2000-2018?). Add this information.

**Response:** "which derived from the similar years in cold-temperate (2003–2016), temperate (2002–2018), subtropical (2000–2017) and tropical zones (2003–2015)." **was added** in Lines 151-153.

**Comment:** Line 149: What do you mean with "winter" in the (sub)tropics? This term does not fit in this context.

**Response:** "winter" **was revised to** "November–April" in Lines 168-169.

**Comment:** Line 154-176: Are the data precise enough to state two decimal places? Confidence in these numbers would be increased if uncertainties were provided.

**Response:** Yes, the annual soil carbon effluxes were precise enough to state two decimal places. With your suggestions, the standard errors were added in Lines 176-202.

**Comment:** Line 154-155: Are these mean annual values averaged across all study sites? What's the considered time period?

**Response:** "Mean annual soil carbon emission was 851.88 g C m$^{-2}$ yr$^{-1}$ in China's forest ecosystems, ranging from 260.10 g C m$^{-2}$ yr$^{-1}$ to 2058.00 g C m$^{-2}$ yr$^{-1}$" **was revised to** "The annual soil carbon effluxes ranged from 260.10 g C m$^{-2}$ yr$^{-1}$ to 2058.00 g C m$^{-2}$ yr$^{-1}$ in China's forest ecosystems, and the mean was 851.88±12.75 g C m$^{-2}$ yr$^{-1}$." in Lines 174-176. The observation period was not considered.

**Comment:** Line 160: These acronyms were not defined before. Please specify.

**Response:** Full names of these acronyms **were added** in Lines 183-185.

**Comment:** Line 165-176: Too many numbers in these paragraphs. I would recommend to provide a comparative figure instead.

**Response:** The comparative figure had been provided in Figure 4, and the comparisons of annual soil carbon emissions among different forest types and climate zones were summarized from Figure 4 in these paragraphs.

**Comment:** Line 177: A general discussion on uncertainties is lacking!

**Response:** "Form Fig. 4 we could also found that the standard errors in tropical and temperate zones (~16 g C m$^{-2}$ yr$^{-1}$) were smaller those in cold-temperate and tropical zones (~65 g C m$^{-2}$ yr$^{-1}$)." **was added** in Lines 254-256.

**Comment:** Line 179-196: A brief comparison with values from other regions outside China could be added.

**Response:** Our results were compared with global forest ecosystems in Line 210.

**Comment:** Line 193: See previous comments regarding the use of R $^{2}$and non-linear models.

**Response:** "The correlations between Rs and soil temperature were lowest in tropical zone (R$^2$=0.225–0.291, Fig. 2d)" **was revised to** "Soil temperature at the depth of 5 cm and 10 cm could only explain 29% and 23% of the Rs variations and RMSEs were 1.09 μmol m$^{-2}$ s$^{-1}$ and 1.13 μmol m$^{-2}$ s$^{-1}$ in tropical zone, respectively (Fig. 2d)" in Lines 218-220.

**Comment:** Line 207-2018: It is difficult to compare any total numbers of Rs (from different sites or studies) if the respective measurement periods, for which these numbers have been computed, are not stated.

**Response:** The annual soil respiration data in these large-scale syntheses were also from the publications. Due to the limit of annual soil respiration data, it is hard to compare the annual means in the same measurement periods. In this paragraph, we compared the mean annual soil carbon efflux with the previous studies, not the total soil carbon efflux.

**Comment:** Line 214: I would suggest to write the full names and regret from using acronyms if the terms are only used a few times throughout the manuscript as in this case. This enhances the readability.

**Response:** The acronyms of forest types were replaced by the full names in Lines 241-246.

**Comment:** Line 237: This number conflicts with the total number of Rs data (=10288) stated at the beginning of the manuscript, doesn't it?

**Response:** A mean or a few values per month were shown in the original papers. The samples of 10288 included monthly means and a few values per month. "monthly means or a few values per month" **was added** in Table 1.

**Comment:** Line 258: This sentence is a bit misleading as no in-situ measurement have been performed. Specify that a comprehensive literature review has been conducted to generate the dataset.

**Response:** "we collected in situ Rs measurements" **was revised to** "we reviewed the Rs-related literatures and collected in situ Rs measurements" in Lines 297-298.

**Comment:** Line 251: Any kind of outlook is missing. Does this new compilation for example indicates that there are particular regions or forest ecosystems that are under-represented with respect to soil respiration and temperature studies and deserve more attention?

**Response:** "It's worth noting that the Rs studies were fewer in the regions of latitude larger than 48 ° (~2%) or elevation higher than 3000 m (~4%). The potentially under-represented forest types might affect the evaluation of temperature sensitivity of soil respiration and annual soil carbon emission at the regional and national scale." **was added** in Lines 287-290.

**Comment:** Figure 1: I would suggest to include more information in this map. A digital elevation model, hillshade, orthophoto or land surface cover classification could be displayed as a background map. Different colours or symbols could be used for the study sites to indicate for example the length of the time series (e.g. 1 year, 5 years, 10 years, 20 years), or the number of available variables at each site (i.e. Rs, T5, T10). The overview map is too small and has no added value. Better increase or remove it. It would also be helpful to indicate the considered climates and/or forest types.

**Response:** Forest types were displayed as a background map, and the average length of the individual time series from the different sites were added in Figure 1.

**Comment:** Figure 2: Note that R ²is invalid/inappropriate for non-linear regressions (see previous comment). Have monthly or annual data been used for the calculation? From which period do the data originate?

**Response:** Root Mean Square Error (RMSE) was added in Figure 2. Monthly soil respiration rates and soil temperature from 2001 to 2018 were used for the calculation.

**Comment:** Figure 4: How and for which period were the mean annual fluxes calculated? I assume all annual data from different years and sites associated with one forest type were spatially and temporally averaged. Is this correct?

**Response:** Yes, all annual data of the same forest type in each climate zone were averaged.

**Comment:** Figure S1: Does the mean reflects the entire study period at each site? Do the correlations look similar if monthly data (collected vs. digitised) were compared?

**Response:** Yes, the mean reflects the entire observation period at each study. The correlations (collected vs. digitized) were excellent and $R^2$ were all larger than 0.99.

**Comment:** Figure S2: I have the impression that there are other non-linear functions that describe the relationship between the soil respiration rate and soil temperature better than the applied ones.

**Response:** Figure S2 showed the relationships of soil respiration rates with soil temperature at 5 cm depth and 10 cm depth across China, Figure 2 showed the respective relationships in four climate zones.

444

**Referee #4:**

**Comment:** Understanding soil respiration in large forest such as in China is of utmost importance, and compiling a database with available data regarding this topic highly relevant for the scientific community and stakeholders. Thus, the topic is worth of publication. However, clarifications should be provided before publication.

In the Introduction, the aims of the paper must be clarified. It seems this will be a review paper but latter the reader find there is data analysis also.

**Response:** Thanks a lot for your comments. "and analyze temperature sensitivity ($Q_{10}$), monthly and annual Rs in cold-temperate, temperate, subtropical and tropical zones." **was added** in Lines 61-63.

**Comment:** In section 2.1, why was the search focused on publications performed 2018?

**Response:** We started from 2019 to collect the related literatures published up to December 31, 2018, it took three years (2019-2021) to established the soil respiration dataset.

**Comment:** L83: it is not clear what you meant. Should this be provided in section 2.3?

**Response:** "Moreover, the data has been carefully cross-checked by the authors and from different sources." **was revised to** "Moreover, the data from the same authors and different sources (e.g. master or Ph. D. dissertation and journal article) has been carefully cross-checked and supplemented.", and moved to section 2.3 in Lines 106-108.

**Comment:** In section 2.3 it is not clear why you used WEBPLOTDIGITIZER. It seems you used it to extrapolate published data. Please, better explain. How much of your database is based on published data and your estimations? Why the selection of 5cm and 10 cm soil depth? Was it linked with data availability?

**Response:** About 82% of the Rs data were extracted from monthly figures with WEBPLOTDIGITIZER, others (e.g., minimum, maximum) were directly given in the original papers (see in Lines 99-101.). Yes, most studies measured soil temperature at 5 cm depth and/or 10 cm depth. "Soil temperature as a main influencing factor, was usually concurrently measured with Rs. Monthly dynamics of Rs and soil temperature at 5 cm depth ($T_5$) and/or 10 cm depth ($T_{10}$) were shown with figures in many literatures." **was supplemented** in Lines 97-99.

**Comment:** L103: I guess there may be other equipment models to perform these measurements, so I suggest to use the name of the equipment (e.g. infrared gas analysers)

**Response:** "Long-term continuous Rs could be monitored with Li-8100 or Li-8150" **was revised to** "Long-term continuous Rs could be monitored with infrared gas analyzers (e.g., Li-8100, Li-8150)" in Lines 110-111.

**Comment:** L105: "The typical days" – does it mean that you did not compile all the available data? (the reader needs to know the aim of the study)

**Response:** No, all the available data were collected and compiled in the database. Soil respiration was usually measured in the typical days with no rain in the original papers. To avoid to misunderstand, " The typical days were usually selected to calculate mean monthly Rs" **was deleted**.

**Comment:** The methodology used for data analysis must be clarified. Did you consider different climatic zones in the analysis? If so, which ones? The methodology used to calculate temperature sensitivity must be explained in section 2.

**Response:** "Monthly and annual Rs were averaged arithmetically in cold-temperate, temperate, subtropical and tropical zones." **was added** in Lines 123-124.

"Temperature sensitivity ($Q_{10}$) is defined as the factor by which Rs is multiplied when temperature increases by 10 ℃ (Davidson and Janssens, 2006; Lloyd and Taylor, 1994), which is usually calculated with the van't Hoff equation (Rs=ae$^{\beta T}$ & $Q_{10}$=e$^{10\beta}$), where Rs is soil respiration rate ($\mu$mol m$^{-2}$ s$^{-1}$), T is temperature ( ℃)." **was added** in Lines 127-130.

**Comment:** L114: how did you estimated this value? Based on a weighted average between emissions and the extent of each climate zone? Please, explain in the methodology.

**Response:** Monthly Rs and annual soil carbon efflux were collected from the original papers, not the estimated values. Monthly Rs and annual soil carbon efflux with other units in some studies were converted to the common unit of $\mu$mol $CO_2$ m$^{-2}$ s$^{-1}$ and g C m$^{-2}$ year$^{-1}$, respectively. "Monthly and annual Rs were averaged arithmetically in cold-temperate, temperate, subtropical and tropical zones." **was added** in Lines 123-124.

**Comment:** L158: how did you assess the significance of the differences? Please, explain it in the methodology

**Response:** With your suggestions, we have added the description of the method in the section "**2.5 Statistical analysis**", "Independent-Samples T Tests (2 groups) and One-Way ANOVA ($\geq$3 groups) at the P = 0.05 significance level were used to test the differences among different forest types in the same climate zone and among the same forest type in different climate zones." **was added** in Lines 124-127.

**Comment:** L160: what is the meaning of EBF, ENF and DNF?

**Response:** " EBF, ENF and DNF" **was revised to** "evergreen broadleaf forest (EBF), evergreen needleleaf forest (ENF) and deciduous needleleaf forest (DNF)" in Lines 183-184.

**Comment:** L162: Meaning of MF?

**Response:** "MF" **was revised to** "Broadleaf and needleleaf mixed forest" in Lines 187-188.

**Comment:** L165: so the analysis was also performed based on ecosystem type? Please, explain it in the methodology

**Response:** "Independent-Samples T Tests (2 groups) and One-Way ANOVA (≥3 groups) at the P = 0.05 significance level were used to test the differences among different forest types in the same climate zone and among the same forest type in different climate zones." **was supplemented** in Lines 124-127.

**Comment:** Section 5: you don't need a separate section to provide this information. You can include it in the Conclusions

**Response:** "Data availability" need to be given as a separate section in Earth system science data.

**Comment:** Fig.1: it will be interesting to show also the climate zones and if possible the main ecosystem types

**Response:** Forest types were displayed as a background map in Figure 1, including evergreen broadleaf forest, deciduous broadleaf forest, evergreen needleleaf forest, deciduous needleleaf forest, broadleaf and needleleaf mixed forest and Bamboo forest.

Thanks again for the reviewers and the editor for your kind consideration and help!

Best regards

Sincerely yours,

Hongru Sun, Zhenzhu Xu, Bingrui Jia

---

## Author Response (AR2)

Title: A compiled soil respiration dataset at different time scales for forest ecosystems across China from 2000 to 2018

**Dear Editors and Referees:**

Thank you very much for your kind consideration and help to our manuscript! According to your suggestions, we furtherly revised our manuscript. All the modifications were listed as follows.

**Referee #1:**

**Comment:** The author basically answer my question well. I recommend to accept it for publication.

**Response:** Thanks again for your useful suggestions to improve our manuscript.

**Referee #3:**

**Comment:** I applaud the authors for the thorough revision of the manuscript and the restructuring of the dataset/repository. Please find my conclusive comments below:

Figure 1: Please use a metric scale bar (meters instead of miles). The small map inlet in the lower right corner has no added value and should be removed. I would suggest to use a different colour for the dots (e.g. grey instead of yellow) so that they are easier to differentiate from the background colours for the forest types.

**Response:** Thank you for your suggestions. We have made some changes in Figure 1. First, we changed the scale bar from miles to kilometers. Second, yes, using yellow points make it indistinguishable from the background, but grey points were also not obvious. So we used dark points. Finally, to best display the China map, its southernmost part (i.e. the small map) is usually put in the lower right corner in most studies (e.g., Feng et al., 2018; Wen et al., 2018; Meng et al.,

2021).

Feng J, Wang J, Song Y, et al. 2018. Patterns of soil respiration and its temperature sensitivity in grassland ecosystems across China. Biogeosciences, 15: 5329-5341.

Meng X, Mao K, Meng F, et al. 2021. A fine-resolution soil moisture dataset for China in

2002-2018. Earth System Science Data, 13: 3239-3261.

Wen JQ, Chuai XW, Li SC, et al. 2018. Spatial-temporal changes of soil respiration across China and the response to land cover and climate change. Sustainability, 10: 4604.

doi:10.3390/su10124604.

**Comment:** I couldn't find the rasterised dataset. Is it included in the ESRI Map Package (.mpg)?

**Response:** We rasterized this dataset to 1 km resolution. Yet compared with the large forest area of

China ($188 \times 10^4 \, \text{km}^2$), the small samples (N=634) and their unbanalanced distributions would result in the large uncertainties. Data coverage is a challenge for larger-scale extrapolation, and the simple interpolation analysis based on this data may be inaccurate. Thus, the rasterized dataset was not included in the manuscript.

**Comment:** ESRI Map Package (.mpg): I appreciate that a file that contains the metadata (coordinates, elevation, study site name, forest type, etc.) for each study site was included. However,

I recommend to provide a geopackage (.gpkg) instead of the proprietary ESRI Map Package as the geopackage can be easily imported in any GIS.

**Response:** The file "metadata.gpkg" was established and updated in the dataset.

**Comment:** When I open the data and metadata files, the hyphen ("�C") is not correctly displayed.

Please check the encoding.

**Response:** I have checked the data and metadata files, the abnormally displayed hyphen " ‒ " was revised to "-".

**Comment:** Line 208: Delete "one"

**Response:** "one" was deleted in Line 196.

**Comment:** Line 267: "Form Fig. 4..." => "From Fig. 4..."

**Response:** "Form Fig. 4" was revised to "From Fig. 4" in Line 254.

**Comment:** Line 269: "...were smaller those..." => "...were smaller than those..."

**Response:** "were smaller those" was revised to "were smaller than those" in Line 255.

Thanks again for the reviewers and the editors for your kind consideration and help!

Best regards

Sincerely yours,

Hongru Sun, Zhenzhu Xu, Bingrui Jia